# A Hybrid Approach for Energy Consumption and Improvement in Sensor Network Lifespan in Wireless Sensor Networks

**DOI:** 10.3390/s24051353

**Published:** 2024-02-20

**Authors:** Arif Ullah, Fawad Salam Khan, Zia Mohy-ud-din, Noman Hassany, Jahan Zeb Gul, Maryam Khan, Woo Young Kim, Youn Cheol Park, Muhammad Muqeet Rehman

**Affiliations:** 1Department of Computer Science, Faculty of Computing and Artificial Intelligent, Air University, Islamabad 44000, Pakistan; arifullah@mail.au.edu.pk; 2Department of Creative Technologies, Faculty of Computing and Artificial Intelligence, Air University, Islamabad 44000, Pakistan; fawad.salam@mail.au.edu.pk; 3Biomedical Engineering Department, Air University, Islamabad 44000, Pakistan; drzia@mail.au.edu.pk (Z.M.-u.-d.); jahanzebgul89@gmail.com (J.Z.G.); 4Department of Software Engineering, Karachi Institute of Economics and Technology (KIET), Karachi 75260, Pakistan; nhassany@kiet.edu.pk; 5Department of Electronic Engineering, Jeju National University, Jeju 63243, Republic of Korea; maryamkhan93@stu.jejunu.ac.kr (M.K.); semigumi@jejunu.ac.kr (W.Y.K.); muqeet1988@jejunu.ac.kr (M.M.R.); 6Department of Mechanical System Engineering, Jeju National University, Jeju 63243, Republic of Korea

**Keywords:** hybrid routing protocol, energy efficiency, network lifetime, hidden Markov model, fuzzy spider monkey

## Abstract

In this paper, we propose an improved clustering algorithm for wireless sensor networks (WSNs) that aims to increase network lifetime and efficiency. We introduce an enhanced fuzzy spider monkey optimization technique and a hidden Markov model-based clustering algorithm for selecting cluster heads. Our approach considers factors such as network cluster head energy, cluster head density, and cluster head position. We also enhance the energy-efficient routing strategy for connecting cluster heads to the base station. Additionally, we introduce a polling control method to improve network performance while maintaining energy efficiency during steady transmission periods. Simulation results demonstrate a 1.2% improvement in network performance using our proposed model.

## 1. Introduction

Sensors, being the cornerstone of modern data acquisition systems, offer unparalleled precision and versatility, enabling advancements in fields ranging from environmental monitoring to healthcare diagnostics through the integration of technologies like MEMS, optical sensing, and wireless communication [1,2]. Wireless sensor networks (WSNs) are a novel type of wireless network that is rapidly gaining traction for both commercial and military applications. A distributed network of sensor nodes makes up the automated network systems. Wireless sensor networks consist of various components that work together to enable data collection, communication, and processing. The primary component of a WSN is the sensor node, which includes a sensing unit to capture data from the environment, a processing unit to analyze and process the collected data, and a communication unit to transmit the data wirelessly to the base station or other nodes in the network. These sensor nodes are typically small, low-power devices equipped with sensors such as temperature, humidity, light, or motion sensors [3]. The base station acts as a central point for data aggregation, storage, and further processing. It serves as the gateway between the WSN and external networks. In addition to sensor nodes and the base station, WSNs may also include sink nodes, which act as intermediate relay nodes to extend the network coverage. For WSN producers, keeping the network operational over an extended period is essential. Because of technological improvements, sensors are now usable in a wide range of industries, including the military, healthcare, transportation, and security [4]. Hence, for the past twenty years, creating energy-efficient protocols has transformed the state of the art. These tiny devices gather data and transmit it throughout the entire network utilizing routing algorithms that primarily make use of wireless sensor network capabilities and ought to be created in a way that maximizes resources in order to create a new probabilistic routing system. In order to process the data generated by sensor networks, new algorithms have been created, and existing data mining approaches have been modified. Numerous knowledge discovery methods, approaches, and algorithms have been put out in the last ten years [5]. By balancing the amount of power utilized by the sensor batteries during the multi-hop data distribution pattern of flat and hierarchical networks, the majority of the created protocols attempted to increase the network’s lifetime [6]. Numerous algorithm methods, such as focusing on classification, sequential patterns, association rules, clustering, and common patterns, have been successful when applied to on-sensor data. Yet due to the massive scale (thousands of sensor nodes), constrained power supply, loss of the communication environment, unsafe deployment, and high failure rate, sensor networks’ design and deployment provide particular research issues. Data collecting is one of the energy-heavy processes, if not the energy-heavy process, that limits the network’s lifetime; it is one of the most important difficulties with WSNs [7]. For that purpose, in this paper, we are going to design our own approach that can improve the network lifetime (in terms of energy, throughput, total energy, and resource utilization in the given network). Figure 1 presents the WSN structure.

## 2. Related Work

Demand-side management (DSM), load forecasting, power pricing laws, energy efficiency initiatives, and customer classification are among the applications that can benefit from the use of consumption pattern knowledge. It is necessary to employ data mining tools to identify consumption patterns. Research has drawn attention to the problem of routing in wireless sensor networks (WSNs). By using a clustering technique and offering a bioinspired ensemble strategy based on the Firefly and SMO algorithms as a clustering-based routing protocol for WSN, the energy in the sensor network can be distributed more fairly [8,9,10]. These protocols lessen the chance of needless energy use by recycling data between the source and sink nodes. Additionally, these protocols can choose the best cluster heads for each round depending on a variety of factors, including intercluster distances to the sink. Using cluster overlaps and node residual energy, the optimal routing path is found. To achieve the optimal outcomes for the network’s requirements, the parameters of the proposed solution can be adaptively changed during the clustering process [11]. Ant and K-means clustering is a unique WSN design that has been created using the colony optimization technique [12]. The fuzzy Dstar-Lite routing technique was used to generate the best information routing for HWSNs. It also elucidates the issue of UEDs within the network and draws attention to the challenge of extending beyond the blockage situation. In [11], a routing scheme for WSNs is put out. It enhances the architecture of the particle swarm algorithm, enabling direct communication between particles as they proliferate, increasing the network’s efficacy. The author [12] suggested choosing the optimal cluster head by use of a genetic algorithm (GA). Four different factors are taken into account when choosing a cluster head (CH) using GA: energy, node density, distance, and mixed nodes’ capacity to build fitness functions. These factors facilitate the determination of the hop count, power capacity, and optimal nodes for CHs within the cluster. The suggested method by [13] increases the longevity of the network and fixes sensor node connection issues. Every node should have a backup route, according to the EFRP, so that sources and destinations can be swiftly relocated. The updated route path can be added to the current one without interruption by employing this method in order to locate and promptly report any oil traces to the washbasin. In [14], a unique ocean surface routing system that integrates two-dimensional underwater sensor networks with sleep scheduling routing was unveiled using a routing solution based on the K-NN algorithm and the clustering method to reduce end-to-end latency and energy consumption [15]. This solution provides the least number of distances through a clustering technique based on node categorization. The authors of [16] made a new contribution to reduce the energy consumption in WSNs’ symmetric routing strategy using two unrelated channels. Every node has two different, shorter paths to the sink in order to reduce network load. Refs. [17,18] described a novel strategy for clustering the HWSNs approach. At this moment, the information bundle receives the chaining technique. The cluster head node, the quantity of sensor nodes, and the remaining energy were all correctly determined using this method. They recommended using the SMORP swarm-based intelligence method in both heterogeneous HWSNs and homogeneous WSNs. Using a set of routing parameters, this method finds the best path for the network [19].

### 2.1. Summary of Related Work

Wireless sensor networks play a crucial role in different fields when we comprehensively review and analyze the existing literature. The summary aims to identify and address the current gaps, limitations, and challenges that exist within the domain of WSNs. By identifying these problems, researchers can gain a deeper understanding of the research landscape and develop innovative solutions to contribute to field issues, such as limited scalability: WSNs are designed to operate in large-scale environments where hundreds or thousands of sensors are deployed. However, scalability remains a significant challenge due to issues such as network management, data aggregation, and energy efficiency [20]. Energy efficiency and power management: energy efficiency is a critical concern in WSNs, as batteries typically power sensor nodes with limited capacity. Prolonging the network lifetime and minimizing energy consumption is essential for successful deployment. The related work section should highlight the existing approaches, protocols, and techniques used to optimize energy consumption and manage power in WSNs. Data security and privacy: WSNs often deal with sensitive data collected from various applications, including environmental monitoring, healthcare, and surveillance. Therefore, ensuring data security and privacy is of utmost importance. The related work section should explore the existing research on security mechanisms, encryption algorithms, authentication protocols, and privacy-preserving techniques for WSNs [21]. Data aggregation and fusion: WSNs generate a vast amount of data, and efficient data aggregation and fusion techniques are necessary to reduce redundancy, conserve energy, and enhance network performance. The related work section should investigate existing methods, algorithms, and protocols for data aggregation and fusion in WSNs, highlighting their advantages, limitations, and potential areas of improvement. Quality of service (QoS) provisioning: WSN applications often require specific quality of service guarantees, such as low latency, reliability, and data accuracy. However, providing QoS in resource-constrained WSN environments presents several challenges [22].

### 2.2. Issue Still Exists

The energy constraint of sensor nodes is a major issue in WSNs since the nodes are battery-powered, and their energy consumption must be minimized to extend the network lifetime. The cluster-based routing protocol is an effective approach to reduce the energy consumption of sensor nodes in WSNs, and the selection of an appropriate cluster head (CH) is a crucial factor in this protocol [23]. This paper proposed a hybrid approach for implementing cluster head selection in WSNs. The proposed approach combines both centralized and distributed approaches and takes into account the network topology, node energy, and residual energy of nodes to select the appropriate cluster head. This paper’s goal is to assess the efficiency of the suggested strategy in reducing energy consumption and prolonging the network lifetime compared to existing approaches.

## 3. WSN Cluster Head Architecture

Numerous restrictions, including scalability, fault tolerance, and energy efficiency, to mention a few, are constantly present and have an impact on the WSN design. By strategically placing sensor nodes, the cluster head selection goal is to determine the minimal transmit power from each node. The edge-bearing sensors will constantly scan their immediate vicinity for nearby sensors to communicate data to while consuming the least amount of transmit power [24]. However, nodes pointed towards network edges often receive complete connectivity from the sensors located between edges. Checking every sensor’s position to verify optimality becomes a computationally difficult process as the network scale increases. To discover the best answer, metaheuristic search strategies are used. Actually, when searching for the best solution—in this case, CH selection and flexible network scaling—there is always a trade-off between accuracy and complexity [25]. The optimization of clustering in wireless sensor networks is shown in Figure 2.

In a clustered architecture, the sensor nodes are methodically organized into clusters, each of which is controlled by a single high-energy CH. Every network cluster’s sensor participates in message transfer across the matching CH, and the CH then transmits the data collected to the BS, which is typically seen as an access point (AP) connected to a wired network. Because of data aggregation and transmission, sensor networks benefit from a clustered network design. Cluster determination in the hierarchical method of network routing is often calculated in relation to the energy retained by the sensors and the geometric proximity of each sensor to the relevant cluster head (CH) [26,27]. While other sensor nodes just transfer their signals to the CH, the CH of each cluster is sufficient to provide all necessary information to the BS. The necessity of maintaining a central node to synchronize every connected node is reduced by clustering. Current wireless sensor networks are always comprised of clusters. When compared to other standard routing techniques, the sensor networks perform well with the aid of clustering, allowing for flexible data exchange and a longer network life [28]. When there are enough sensor nodes, WSNs may be organized on an as-needed basis. The routing table’s dimension decreases because clustering contributes to the maintenance of communication bandwidth. The requirement to maintain the existing network topology is removed by clustering. An energy-efficient clustered WSN uses less energy overall. Every sensor node’s battery life is increased because of the network’s predicting behavior, and in the event that clustering is done correctly, upscaling of the network is conceivable. The size of the clusters, intracluster contact, mobility of the sensors and cluster heads, sensor variety and location, various levels, and overlaps are the main design factors taken into account while building up a network clustering [29]. Connectivity, rotating cluster head functions, medium access control layer drawing, sensor duty cycle, optimal cluster dimension, and sensor harmony with peer nodes are some of the major clustering issues. Every time a node is moved from one to the other, the CH accumulated data are updated. The process of choosing the best communication path for data packets to take from a source node to a destination node is known as path selection in a wireless sensor network (WSN). In WSNs, where sensor nodes are frequently resource-constrained in terms of energy, computing power, and communication range, this technique is essential for effective and dependable data transmission [30,31]. Figure 3 presents the path selection in a WSN.

The process of choosing a path entails figuring out which intermediary nodes the data packets will pass through in order to reach their final destination. To guarantee efficient communication and low energy usage, the chosen channel should take into account a variety of criteria [32]. Given that WSNs are frequently used in vital applications, including healthcare, industrial monitoring, and military surveillance, security is an essential component of these networks. The are some typical WSN security problems: data confidentiality, data integrity, authentication, energy efficiency, scalability, and location privacy; for the improvement of security, we can contribute to this section.

### Mathematical Model of Path Selection in WSN

In wireless sensor networks (WSNs), mathematical models for path selection are frequently created to optimize a variety of goals, such as energy efficiency, latency reduction, or dependability. Graph theory and optimization methods are frequently used in these models. A condensed mathematical model that illustrates the process of path selection based on the fundamental goal of minimizing overall energy usage in a WSN is provided in [33]. G=(V,E) represents the communication graph of the WSN, where V is the set of sensor nodes and E is the set of communication links between nodes. s denotes the source node, d denotes the destination node, P represents the selected path from s to d, $E_{n}$ is the energy level of node n, $C_{n}$ is the energy consumption rate of node n when transmitting data, $L_{ij}$ represents the link quality or reliability between nodes i and j. D represents the data rate. Objective: minimize the total energy consumption along the selected path P. Constraints: connectivity; ensure that the selected path P is connected in the communication graph G energy constraint [34]. The energy consumption along the path should not exceed the energy available at any node. Mathematically, for each node n along path P, the following constraint should hold:(1)$$\sum_{i\in P}C_{i}\leq E_{n}$$

Optimization problem: formally, the problem can be represented as an optimization problem: minimize [30].
(2)$$\sum_{i\in P}C_{i}$$

Subject to s to d connectivity constraint. Energy constraint for each node n along path P. Mathematical model: depending on the particular needs and limitations of the WSN, this optimization issue may be tackled using linear programming, integer programming, or other optimization approaches. The goal is to identify path *P* that satisfies the connection and energy limitations while minimizing overall energy use. It is vital to keep in mind that this is a mathematical model that has been simplified and that in real-world path selection in WSNs, more complicated factors like routing protocols, changing network circumstances, and QoS need to frequently come into play [31]. The mathematical model may need to be adjusted or modified based on the unique application and goals. Master node selection: a master node selection is the process of selecting one particular sensor node among the sensor nodes in a wireless sensor network (WSN) to act as the master or central node. The network and other sensor nodes’ operations are often managed and coordinated in large part by the master node. The network’s overall effectiveness and efficiency may be significantly impacted by the choice of a master node [32].

## 4. Proposed Model

In this section, the proposed framework of fuzzy spider monkey optimization (FSMO) and a hidden Markov model (HMM) are briefly introduced. Every protocol round has two distinct phases: setup and steady state. During setup, the CH selection process is simplified. BS uses SMO as a tool during the setup phase to construct energy-efficient clusters for a specific NAN sensor, network remaining energy, and no overlapping distance. During the steady-state phase, the CHs collect data from the individuals in their own cluster and transmit it to a base station (BS) [33]. Figure 4 presents the proposed model architecture.

As we know, there are different techniques used for the selection of a routing path in a WSN, but for our approach, we used the multipath routing approach; the details for the approach are mentioned and discussed in the next section. A mathematical model of a hidden Markov model (HMM) describes the probabilistic linkages and transitions between hidden states and observable data in a wireless sensor network (WSN). In WSNs, temporal data like sensor readings, outside circumstances, or network states are frequently modeled using HMMs. An HMM in the context of a WSN is represented mathematically in the following manner.

State space:

Hidden states: $Q=\{q_{1},q_{2},\ldots ,q_{N}\}$, where $q_{i}$ represents the i-th hidden state.

Observation space:

Observed symbols: $O=\{o_{1},o_{2},\ldots ,o_{T}\}$, where $o_{t}$ represents the t-th hidden state.

Model parameters:

Initial state distribution: $\pi =[\pi_{1},\pi_{2},\ldots ,\pi_{N}]$, where $\pi_{i}$ is the probability of starting in state $q_{i}$.

State transition probability matrix: $A=[a_{ij}]$, where $a_{ij}$ is the probability of transitioning from state $q_{i}$ to state $q_{j}$.

Observation probability matrix: $B=[b_{j}(k)]$, where $b_{j}(k)$ is the probability of observing symbol k from hidden state $q_{j}$ [35].

HMM components:

State transition probabilities:$${\;a}_{ij}=P\left(q_{t}+q_{j}\right|q_{t-1}=q_{i})\;\;for\;\;1\leq i,j\leq N$$

Observation probabilities:$$b_{j}(k)=P\left(o_{t}=k\right|q_{t}=q_{j})\;\;for\;\;1\leq j\leq N,1\leq k\leq M$$

Forward algorithm: computes the probability of observing a sequence O given the HMM parameters.

Viterbi algorithm: finds the most likely sequence of hidden states Q given the observed sequence, O.

Baum–Welch algorithm: estimates the HMM parameters, π, A, and B from training data.

Utilizing the SMO approach, the network lifespan is improved. If a node is unable to send data owing to damage, work with surrounding nodes to replace it. By employing node replacement, the cluster head SMO version disclosed in this study enhances the performance of the prior SMO. The challenge of keeping them contained in a small space led to the development of the spider monkey strategy. The mathematical model for SMO is given by Equation (3).
(3)$$T_{y}^{x}=\left\{\begin{array}{c} ET_{y}+q_{1}[\left(VC_{y}-NC_{y}\right)q_{2}+NC_{y}]\;\;q_{3}\geq 0\\ ET_{y}-q_{1}[\left(VC_{y}-NC_{y}\right)q_{2}+NC_{y}]\;\;q_{3}<0 \end{array}\right.$$
where $q_{1},q_{2}$ are random integers based on the interval [0, 1], $NC_{y}$ is the upper bound in the *y*th dimension, $ET_{y}$ is the position of the food source in the *y*th dimension, and $T_{y}^{x}$ is the first cluster head position in the *y*th dimension. The most important component is the significant coefficient *q*_1_, which is used in Equation (4) to balance the processes of food acquisition and consumption.
(4)$$q_{1}={2f}^{{-(\frac{4m}{M})}^{2}}$$

The most recent round is denoted by *L*, the most rounds by *M*, and the significant SSA coefficient is denoted by $q_{1}$. The generated routing route is utilized frequently (rounds) in FSMO, and each node’s state is assessed along the way to determine whether to use the same path for the subsequent round. The prior assumption states that the sink has access to up-to-date data on the battery life, position, and network traffic load of every node. The fitness of a contiguous node ${(n}_{i})$ can be found using the following formula [33].
(5)$$fit\left(n_{i}\right)=fuzzy(RE\left(n_{i}\right),TL\left(n_{i}\right),D\left(n_{i}\right))$$

The residual energy, traffic load, and distance to the destination for node *n*, denoted as $RE\left(n\right),TL\left(n\right),\;and\;D\left(n\right)$ accordingly, are the inputs to the fuzzy technique that will determine the node n’s fitness value. The GLSM then evaluates the data obtained from each of LLSM’s neighbor nodes and selects the best node with the highest probability *P* and the given probability value:(6)$$P\left(n_{i}\right)=\frac{fit\left(n_{i}\right)}{\sum_{i=1}^{n}fit\left(n_{i}\right)}$$

N is the number of neighbor nodes, $n_{i}$, $fit\left(n_{i}\right)$ is the fitness associated with node *n*, and $P\left(n_{i}\right)$ is the probability associated with node n. A fuzzy inference engine processes all of these rules concurrently. The solution fuzzy space is reduced to a single, clean output value using defuzzification. This figure represents the fitness function value of node *s* [35].
(7)$$fit\left(n\right)=\frac{\sum_{i=1}^{n}U_{i}*C_{i}}{\sum_{k=1}^{n}U_{k}}$$

Six steps make up the scientific model of SMO’s search behavior for optimization jobs. When spider monkey populations are first created, SMO randomly creates colonies of them. Spider monkeys are represented as D-dimensional vectors. Let $Q_{ab}$ represent the *b*th dimension of each person. The initialization of each $Q_{ab}$ in spider monkey optimization is as follows:(8)$$Q_{ab}=Q_{minb}+S\left(0,1\right)\times (Q_{maxb}-Q_{minb})$$
where $Q_{minb}$ and $Q_{maxb}$ are upper and lower bounds in bth direction for $Q_{a}$ and *S* (0, 1) indicates a random amount between the range [0, 1]. Initialization stage: the Bernoulli procedure is employed in the first phase of the SMO method to randomly initialize a population of N spider monkeys (SM) [36].
(9)$${SMO}_{u,v}=\left\{\begin{array}{c} 1,\;\;\;\;\;\;\;\;\;a<prob\\ 0,\;\;\;\;\;\;\;\;\;otherwise \end{array}\right.$$
where ${SMO}_{u,v}$ is the vth dimension of uth spider monkey, a random number distributed uniformly within the interval [0, 1], and *prob*, a probability with a value of 0.5. The appropriateness ${SMO}_{u}$ of a randomly generated solution (for the minimization problems) is assessed as follows:(10)$${fitness}_{u}=\left\{\begin{array}{c} 1+\left|f_{u}\right|,\;\;F_{u}\leq 0,\\ \frac{1}{1+F_{u}},\;\;F_{u}\geq 0 \end{array}\right.$$
where ${fitness}_{u}$ is the issue under consideration’s fitness function stage of the local leader. Stage two entails revising the solution in light of the team’s and the local leader’s experiences. A binary optimization problem has been addressed using the logical OR, AND, and XOR operators. Each SM changes its position or velocity update equation in the third phase using the knowledge that the group leader and other members have [37]. Hence, the given section reduces the distance from the source node to the destination and helps in the path selection of the master node in the WSN. The cluster head selection method works dynamically because the process needs to be continuous in the network. Clustering is typically achieved in WSNs with the combination of different sections, which are initialization, node election, cluster head formation, cluster formation, and communication setup. These steps are discussed with mathematical representation as in the above section with details.

## 5. Simulation and Evaluation Parameters

The tests are run to gauge how well the suggested algorithms work. On the Anaconda (Spyder) IDE, the simulations are programmed in the Python language. The tests were performed on a computer with an Intel(R) Core (TM) i7 8700 @3.20 GHz processor clocked at 3.19 GHz and 16 GB RAM. Time spent missing deadlines, makespan, energy utilized, overall cost, and degree of imbalance (DI) are the measures used to assess performance. Table 1 presents the simulation parameters.

## 6. Results and Discussion

Table 2 provides these circumstances specifics. As shown in Table 3, the suggested strategy was put to the test in five different scenarios with varying network area sizes, grid/cluster counts, and total node counts. In these scenarios, the number of grids varies depending on the case and ranges from 8 to 40, while the node population varies depending on the network area size, ranging from 100 to 400.

Figure 5 and Figure 6 present the comparison data of the network lifetime after the findings spanning different node densities and network grids were studied. The comparative measures were first node dead (FND), half node dead (HND), and final node dead (LND). Based on the given result, the worst case was the FPS-R algorithm, and the best case was the proposed algorithm. The overall result of the network lifetime is mentioned in Table 3 with details, and Table 4 presents the result of throughput in terms of megabits per second (Mbps), which is calculated for the initial time to finish for the network time.

Figure 7 and Figure 8 present the comparison data of the throughput in the given network in terms of Mbps. The measure was conducted from source to destination. Based on the given result, the worst case was the FPS-R algorithm, and the best case was the proposed algorithm. The overall result is that 0.9867% of nodes are missing to reach the destination. In the worst case, 0.9990% is taken by the FPS-R algorithm. The details of the result are mentioned in Table 4 and Table 5, which present the result of energy consumption/Mj in the network.

The BS was positioned at the network’s edge, as seen in Figure 9, Figure 10, Figure 11, Figure 12, Figure 13, Figure 14 and Figure 15, in order to finish the investigation and assess the effectiveness of the suggested method. This study made it possible to assess the proposed model performance when the base station is situated on the edge of a configuration with 100–400 nodes. From Figure 6 and Figure 9, different types of energy utilization are measured, such as energy use by node and total node in the cluster. The proposed model in this study also showed noteworthy gains in a number of performance indicators.

The substantial increase in the quantity of packets transmitted to the BS is shown in Figure 9, Figure 10, Figure 11 and Figure 12 and Table 3 and Table 4, with an improvement of 3.34 death nodes and a 97% alive node ratio, respectively. Figure 12 and Figure 15 illustrate how the proposed model performed in terms of packet delivery ratio. The details of the given result are mentioned in Table 5 and Table 6 with details. The proposed model improves the network in terms of throughput, energy, packet loss, and active node based on the result of a 1.2% improvement in the result as compared with the standard and research proposed algorithm.

## 7. Conclusions

The aim of the study was to create a hybrid model based on the FSMO and HMM models in order to increase the network lifetime in a WSN environment. In this technique, the optimum choice of the cluster head (CH) is enabled by a trust model among the stationary nodes after network construction, enabling reliable data broadcast from the sensor nodes to the CH. The proposed method of building simulation environments and conducting five separate tests with random pathways and node sizes ranging from 100 to 400 helps the optimization algorithm identify the best methods for delivering data packets to the sink node as quickly as possible. It is possible to select the optimal routes from the random roads by considering factors like route score, total hop counts, residual energy, power used, and the number of received and delayed packets. Because these routes can reliably transfer packets, higher residual energy can be achieved. In all five case studies, the suggested model led to longer network lifetimes than the present proposed model method, which involves faster route selection. The fact that, after several iterations, the percentage of dead nodes in the network was far lower than that of the alternative protocol indicates how successful the current method is. The current effort aims to develop the work that has already been provided in the direction of data reduction in conjunction with security in order to further enhance network efficiency and privacy. Deep learning algorithms may be used in the future to create bufferless systems that can process incoming data from several IoT devices at a speed that matches. In addition, a cross-layer strategy will be investigated in future studies to improve network lifespan and efficiency in the event of node failures.

## Figures and Tables

**Figure 1 sensors-24-01353-f001:**
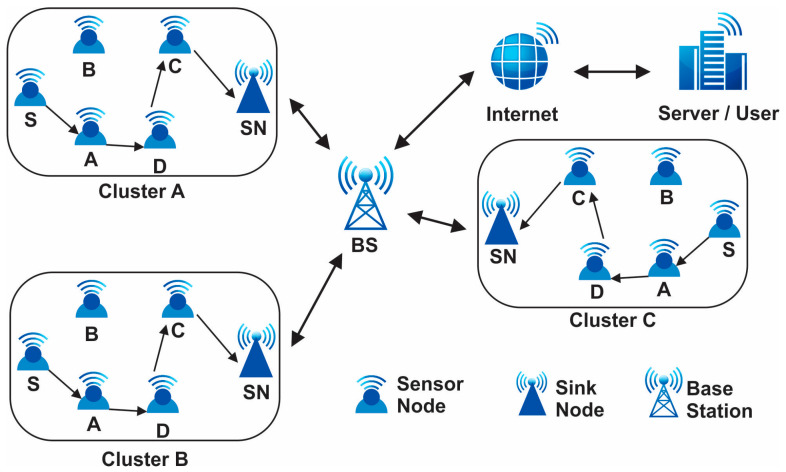
Wireless sensor networks architecture.

**Figure 2 sensors-24-01353-f002:**
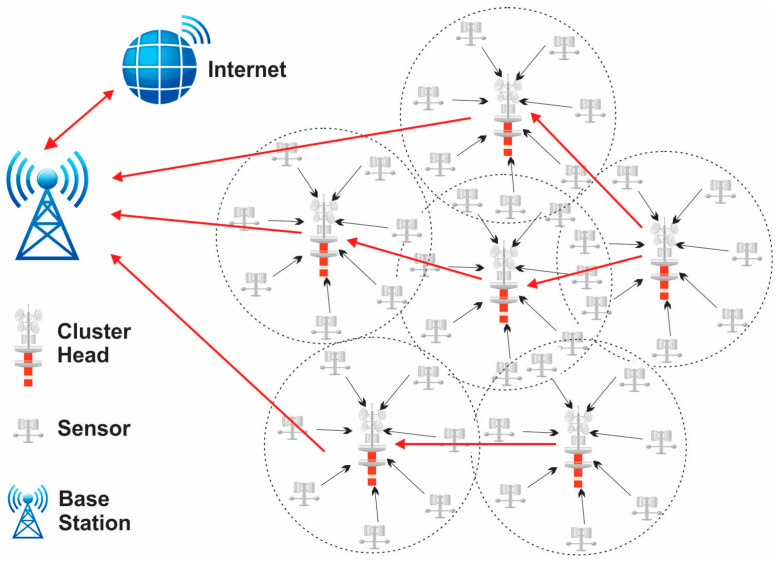
Optimization of clustering in wireless sensor networks.

**Figure 3 sensors-24-01353-f003:**
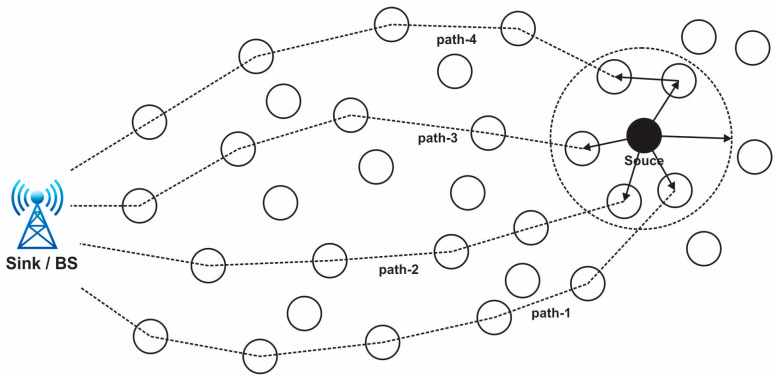
View of the path selection in a wireless sensor network.

**Figure 4 sensors-24-01353-f004:**
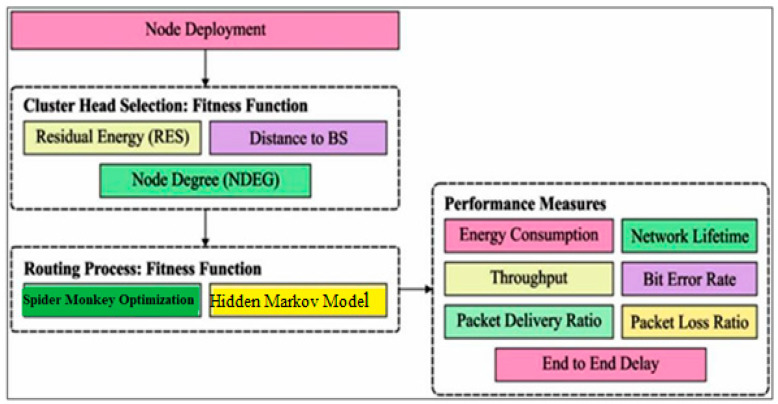
Proposed model architecture.

**Figure 5 sensors-24-01353-f005:**
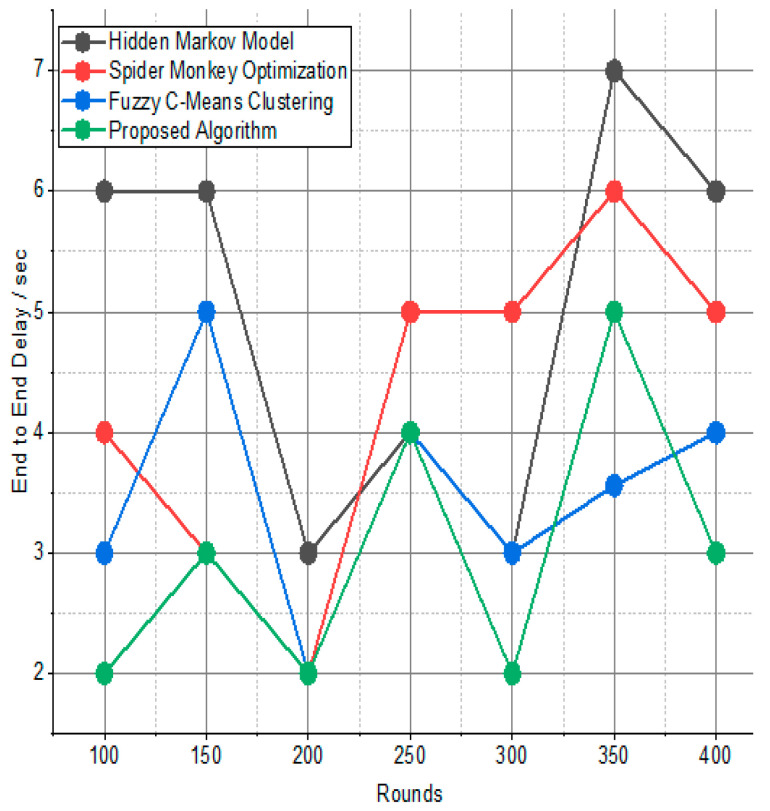
Comparison of alive nodes.

**Figure 6 sensors-24-01353-f006:**
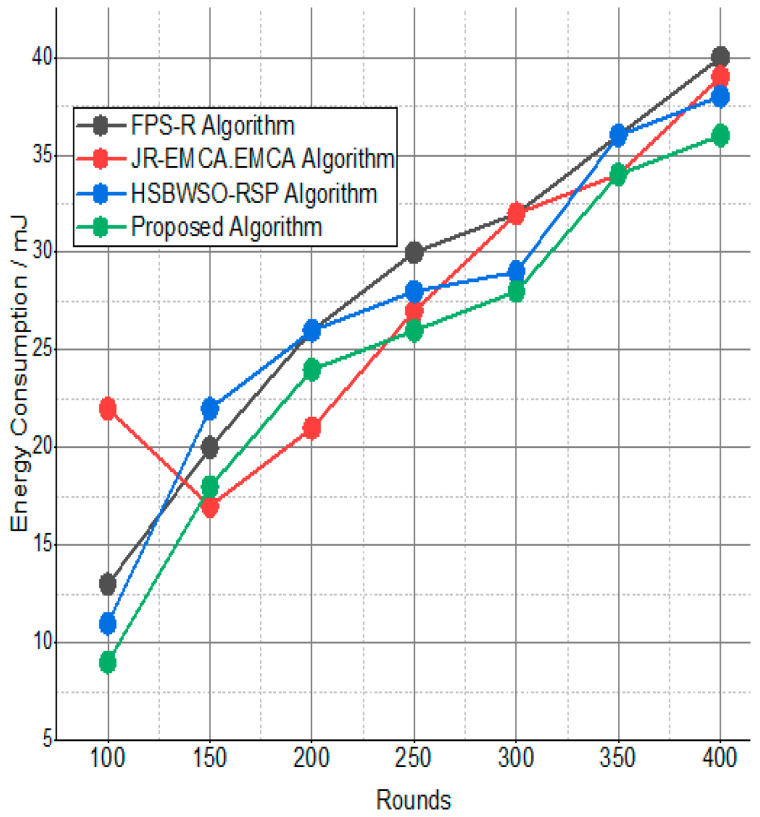
Network lifetime based on rounds.

**Figure 7 sensors-24-01353-f007:**
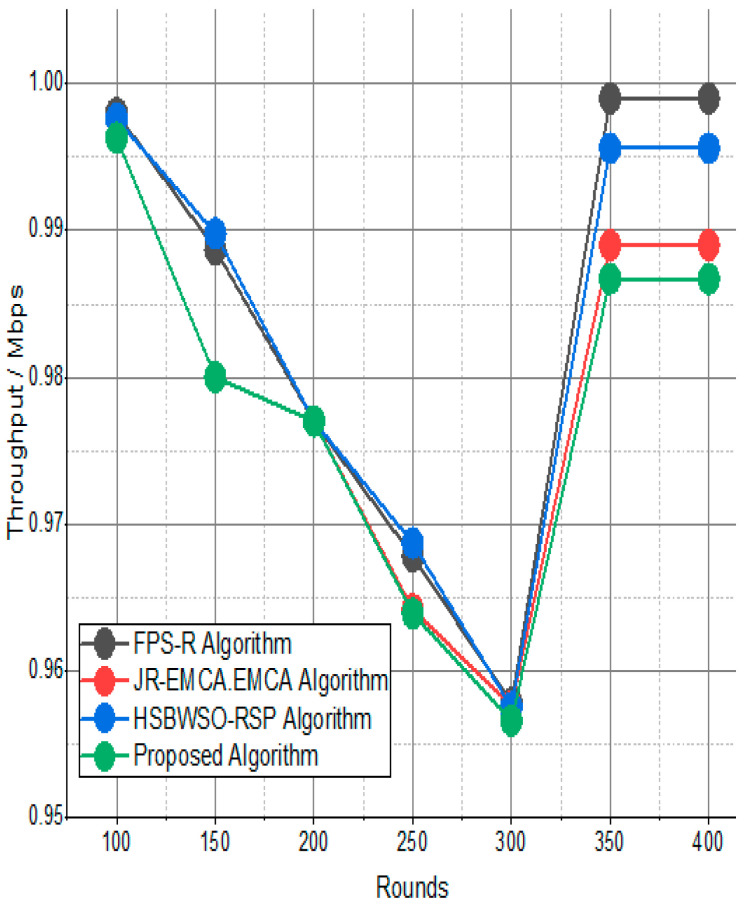
Network throughput/Mbps.

**Figure 8 sensors-24-01353-f008:**
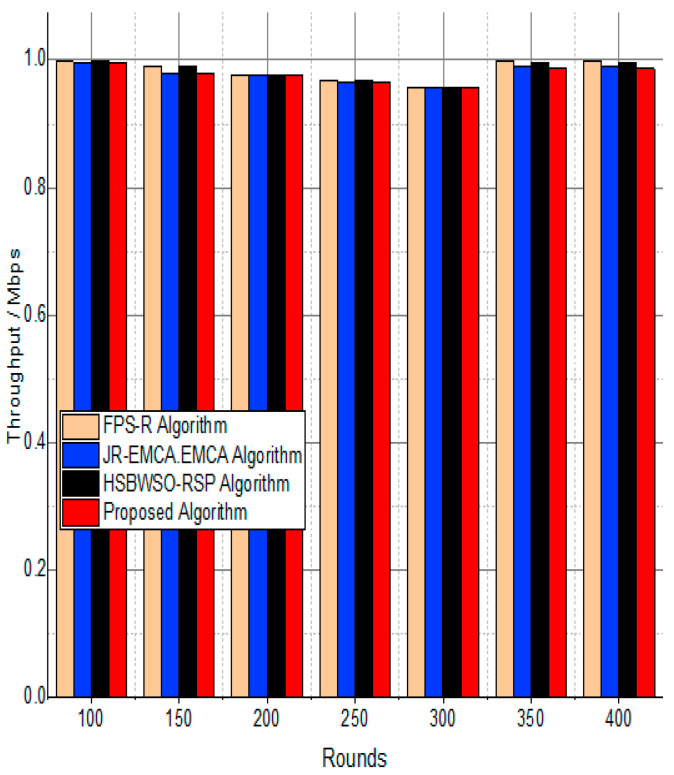
Throughput/Mbps.

**Figure 9 sensors-24-01353-f009:**
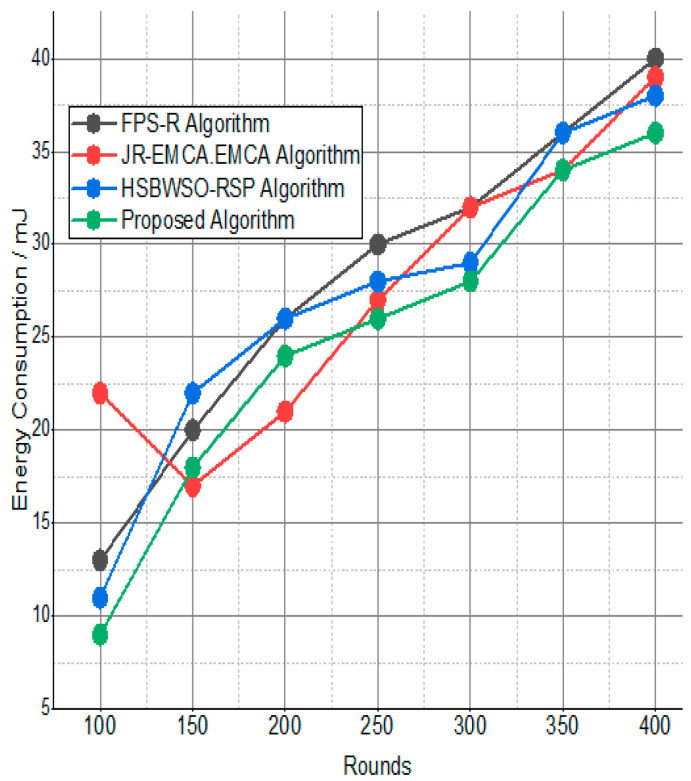
Energy consumption/mJ.

**Figure 10 sensors-24-01353-f010:**
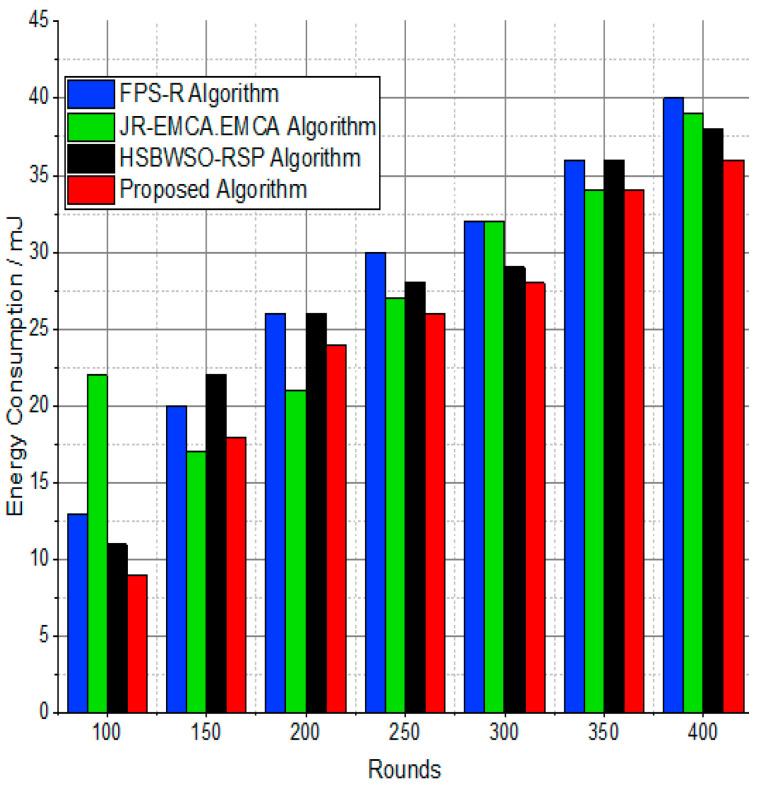
Energy consumption/mJ.

**Figure 11 sensors-24-01353-f011:**
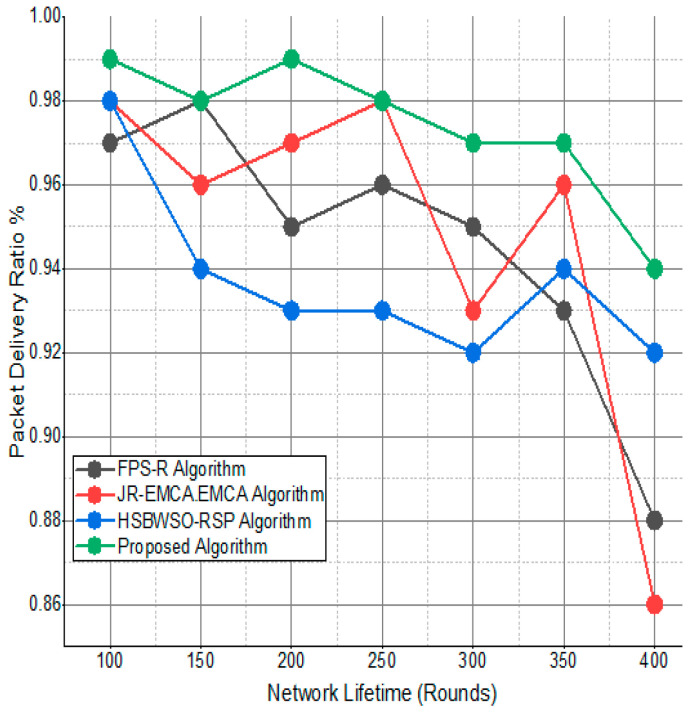
Packet delivery ratio/%.

**Figure 12 sensors-24-01353-f012:**
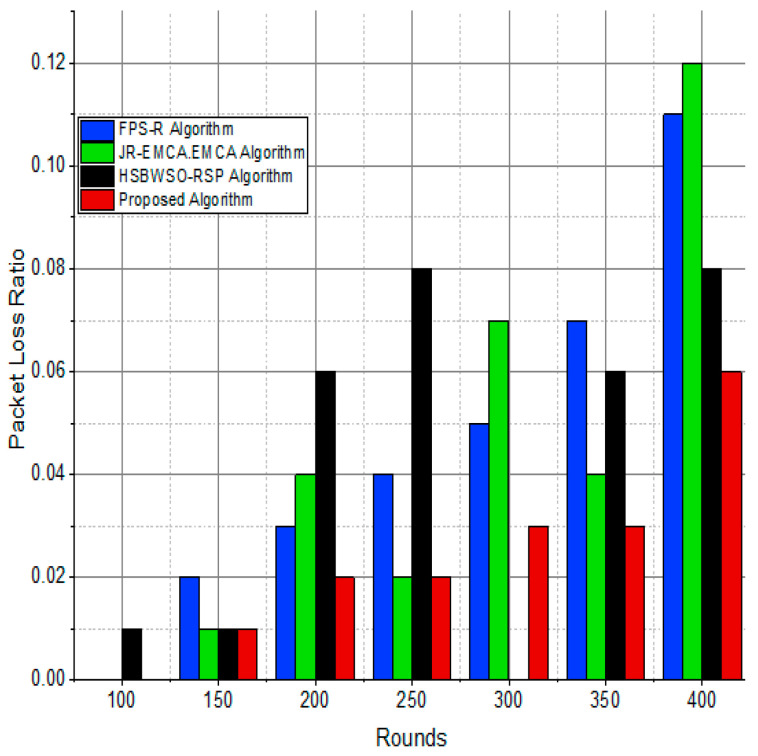
Packet delivery ratio/%.

**Figure 13 sensors-24-01353-f013:**
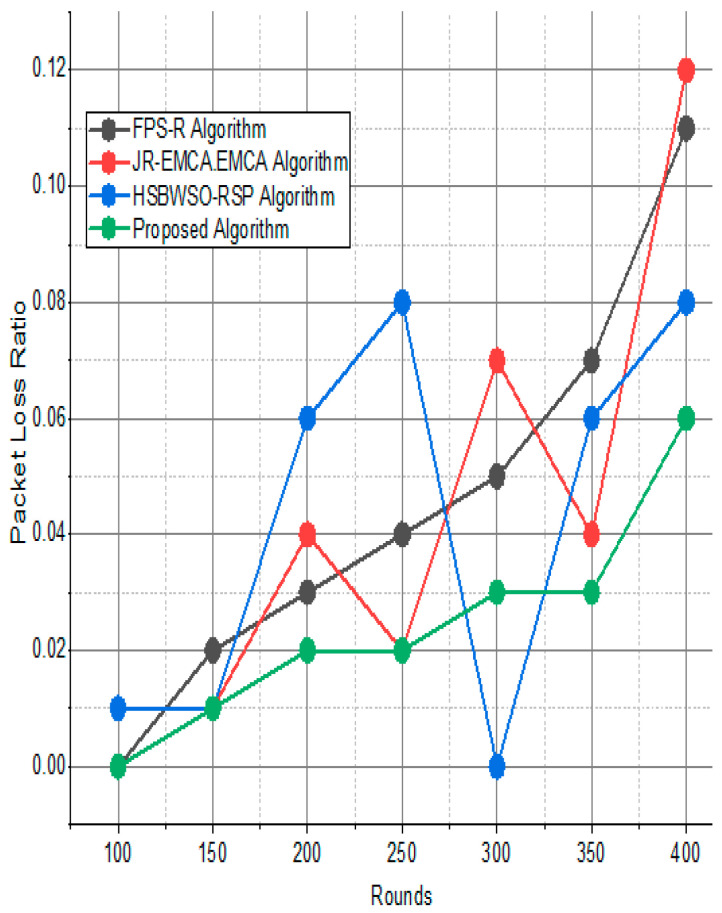
Network packet loss ratio/%.

**Figure 14 sensors-24-01353-f014:**
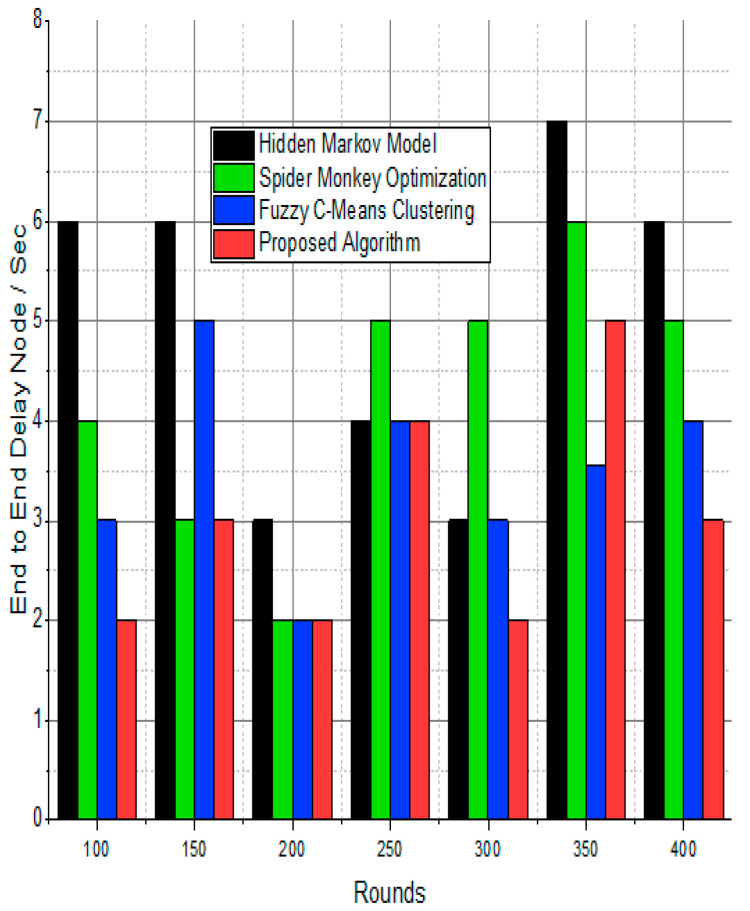
Node delay ratio/%.

**Figure 15 sensors-24-01353-f015:**
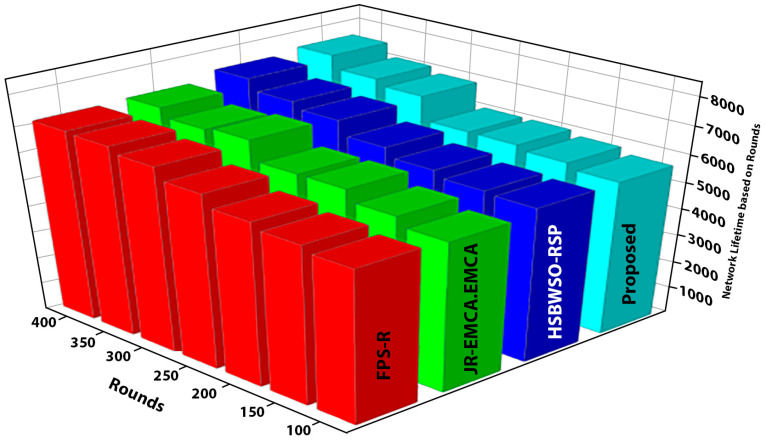
Overall network performance.

**Table 1 sensors-24-01353-t001:** Parameters simulation.

Parameters	Measurements	Parameters	Measurements
Number of Nodes	50–100	Packet Size	1024 bits
Area Size	100 m × 100 m	Communication Range	30 m
Base Station Coordinate	50, 50	Buffer Size	20 packets
Communication Range	30 m	Data rate	4096 bit/round
Cache queue length	50 packets	Traffic Pattern	Constant Bit Rate

**Table 2 sensors-24-01353-t002:** Comparison of alive nodes.

Round	Number of Alive Nodes in First Simulation	Number of Alive Nodes in Second Simulation	Number of Alive Nodes in Third Simulation
100	100	100	100
150	89	87	80
200	67	65	70
250	56	59	50
300	40	58	30
350	23	15	10
400	5	7	0

**Table 3 sensors-24-01353-t003:** Network lifetime based on rounds.

Rounds	FPS-R Algorithm	JR-EMCA.EMCA Algorithm	HSBWSO-RSP Algorithm	Proposed Algorithm
100	5600	5500	5700	5710
150	5800	5810	5780	5900
200	6000	62,000	5990	6050
250	6400	6200	6300	6050
300	6800	6890	6789	6889
350	7000	6789	7010	7089
400	7050	7100	7400	7550

**Table 4 sensors-24-01353-t004:** Throughput/Mbps.

Rounds	FPS-R Algorithm	JR-EMCA.EMCA	HSBWSO-RSP	Proposed Algorithm
100	0.9980	0.9963	0.9976	0.9963
150	0.9887	0.9800	0.9898	0.9800
200	0.9770	0.9770	0.9770	0.9770
250	0.9678	0.9642	0.9687	0.9639
300	0.9578	0.9576	0.9575	0.9566
350	0.9990	0.9890	0.9956	0.9867
400	0.9990	0.9890	0.9956	0.9867

**Table 5 sensors-24-01353-t005:** Energy consumption/mJ.

Rounds	FPS-R Algorithm	JR-EMCA.EMC	HSBWSO-RSP	Proposed Algorithm
100	13	22	11	11
150	20	17	22	20
200	26	21	26	26
250	30	27	28	28
300	32	32	29	30
350	36	34	36	36
400	40	39	38	36

**Table 6 sensors-24-01353-t006:** Packet delivery ratio/%.

Network Lifetime (Rounds)	FPS-R	JR-EMCA.EMCA	HSBWSO-RSP	Proposed Algorithm
100	97%	98%	98%	99%
150	98%	96%	94%	98%
200	95%	97%	93%	99%
250	96%	98%	93%	98%
300	95%	93%	92%	97%
350	93%	96%	94%	97%
400	88%	86%	92%	94%

## Data Availability

Data are contained within the article.

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
