# Peer review of "A Hybrid Approach for Energy Consumption and Improvement in Sensor Network Lifespan in Wireless Sensor Networks"

_sensors, 2024, doi:10.3390/s24051353_

Round 1

Reviewer 1 Report

Comments and Suggestions for Authors
  1. The terminology used in the paper seems outdated; consider updating it to reflect current trends and standards in wireless sensor networks.

  2.  
  3. The paper lacks references to recent advancements and breakthroughs in the field of wireless sensor networks. It is essential to provide a comprehensive literature review to contextualize the proposed technique.

  4.  
  5. The abstract is too lengthy and detailed; try to condense the information to provide a more concise overview of the paper's contribution.

  6.  
  7. The paper doesn't discuss potential limitations or challenges associated with the proposed technique, which is crucial for a well-rounded scientific analysis.

  8.  
  9. The methodology section could benefit from a more detailed explanation of the experimental setup and data collection process. This information is crucial for the reproducibility and validation of the proposed approach.

  10.  
  11. The language used in the paper lacks precision and clarity; consider revising to improve the overall readability and understanding of the content.

  12.  
  13. The paper should explore the implications of the proposed technique in real-world scenarios. Discuss how it can be applied practically and its potential impact on existing WSN deployments.

  14.  
  15. The conclusion section is brief and could be strengthened by summarizing the key findings and emphasizing the significance of the proposed model in the broader context of wireless sensor networks.

  16.  
  17. The paper does not address the issue of security in wireless sensor networks, which is a critical aspect in contemporary research. Consider incorporating a discussion on security considerations and potential threats.

  18.  
  19. The results section could benefit from a more in-depth analysis and interpretation of the simulation results. Provide insights into the practical implications of the observed improvements and how they contribute to the advancement of the field.

Reviewer 2 Report

Comments and Suggestions for Authors

1. Is the cluster head selection method in this paper only for static networks

2. What kind of reasons will cause nodes to exit the network? Is this paper only considering the possibility of energy exhaustion?

3. Suggest the author to check if there are any writing issues with the expression of aij in Line 285, and if there are any writing issues with the expression of Line 308?

4. We don’t see r1 in the Eq(4) for Line 306, which is the most important component mentioned in paper.

5. There are multiple writing issues in this paper. For example, “sequenceO” in Line291 should be “sequence O”, “parameters𝜋” in Line 292 should be “parameters 𝜋”. Suggest the author to carefully review and modify to avoid writing issues.

6. What is the basis for using the value of fitness to define probability P in Line 320?

7. What is the difference between Eq(5) and Eq(7)? What does Table.2 want to reflect? Suggest adjusting the content layout of Table.2 and Table.3.

8. What does Throughput explain and how can it reflect the advantages of this algorithm?

9. Figure.6 and Figure.9 are similar. It is recommended that all images be of the same size as possible, neither large nor small. Pay attention to the size and position of the icon examples in the images for aesthetics.

10. In the section of conclusion, it is appropriate to combine one's own experimental results to summarize and look forward to the future. Meanwhile, suggest the author to check whether the format of the reference meets the template requirements.

Reviewer 3 Report

Comments and Suggestions for Authors

This paper proposes a cluster head selection scheme by fuzzy technique to extend the network life time for wireless sensor networks.

Some comments about this paper are listed as follows.

1.      Cluster head selection scheme is an old issue for wireless sensor networks. There are many solutions have been published to extend the network life time. The authors should update the reference papers to enhance the motivation of the paper.  

2.      It is not clear how to form a cluster.

3.      It is also not clear how to discover a routing path.

4.      The authors say that “the proposed approach combines both 161 centralized and distributed approaches and takes into account the network topology, 162 node energy, and residual energy of nodes to select the appropriate cluster head”. However, I cannot find the distributed scheme.

Comments on the Quality of English Language

The quality of the English language should be improved.

Round 2

Reviewer 1 Report

Comments and Suggestions for Authors

All comments are rectified. Please accept in present form.